

# Effects of miniaturization in the anatomy of the minute springtail *Mesaphorura sylvatica* (Hexapoda: Collembola: Tullbergiidae)

Irina V. Panina[1], Mikhail B. Potapov[2,3] and Alexey A. Polilov[1]

[1] Department of Entomology, Faculty of Biology, Moscow State University, Moscow, Russia
[2] Department of Zoology and Ecology, Institute of Biology and Chemistry, Moscow State Pedagogical University, Moscow, Russia
[3] Senckenberg Museum of Natural History Görlitz, Görlitz, Germany

Corresponding authors
Irina V. Panina,
i.vl.panina@gmail.com
Alexey A. Polilov,
polilov@mail.bio.msu.ru,
polilov@gmail.com

## ABSTRACT

Smaller animals display pecular characteristics related to their small body size, and miniaturization has recently been intensely studied in insects, but not in other arthropods. Collembola, or springtails, are abundant soil microarthropods and form one of the four basal groups of hexapods. Many of them are notably smaller than 1 mm long, which makes them a good model for studying miniaturization effects in arthropods. In this study we analyze for the first time the anatomy of the minute springtail *Mesaphorura sylvatica* (body length 400 μm). It is described using light and scanning electron microscopy and 3D computer reconstruction. Possible effects of miniaturization are revealed based on a comparative analysis of data from this study and from studies on the anatomy of larger collembolans. Despite the extremely small size of *M. sylvatica*, some organ systems, e.g., muscular and digestive, remain complex. On the other hand, the nervous system displays considerable changes. The brain has two pairs of apertures with three pairs of muscles running through them, and all ganglia are shifted posteriad by one segment. The relative volumes of the skeleton, brain, and musculature are smaller than those of most microinsects, while the relative volumes of other systems are greater than or the same as in most microinsects. Comparison of the effects of miniaturization in collembolans with those of insects has shown that most of the miniaturization-related features of *M. sylvatica* have also been found in microinsects (shift of the brain into the prothorax, absent heart, absence of midgut musculature, etc.), but also has revealed unique features (brain with two apertures and three pairs of muscles going through them), which have not been described before.

## INTRODUCTION

Miniaturization plays an important role in morphological changes in animals and has become a popular area of research (e.g., *Hanken & Wake, 1993*; *Polilov, 2016a*). Many arthropods are comparable in size with unicellular organisms and are of great interest for studying miniaturization in animals. Miniaturization implies major morphological

changes of structures and is often accompanied by allometric changes in many organs (*Polilov, 2015a*). The tremendous size changes that occurred in the evolutionary history of arthropods may have allowed them to occupy a vast range of niches. Studies on miniaturization in different arthropods can help us understand what limits body size in organisms and how miniaturized organisms evolved.

Morphological traits (rev.: *Polilov, 2015a*; *Polilov, 2016a*; *Minelli & Fusco, 2019*), scaling of organs (*Polilov & Makarova, 2017*), and even cognitive abilities (*Van der Woude, Martinus & Smid, 2018*; *Polilov, Makarova & Kolesnikova, 2019*) associated with miniaturization have been studied in insects. Studies on other minute Panarthropoda are scarce (*Eberhard & Wcislo, 2011*; *Dunlop, 2019*; *Gross et al., 2019*).

Studies on the miniaturization of insects and the anatomy of the smallest insects (adult body length smaller than 2 mm) show significant changes in the anatomy of microinsects correlated with their size. Some changes are commonly shared by remotely related taxa, and seem to be straightforward adaptions to physical constrains, e.g., the reduction of circulatory and tracheal systems, absence of midgut musculature; compactization, oligomerization, and asymmetry of central nervous system (*Polilov, 2016a*). However, some microinsect taxa possess their own original modifications, such as the complete shift of the brain into the thorax in the adult (*Polilov & Beutel, 2010*) in the beetle *Mikado* sp., or the lysis of cell bodies and nuclei of neurons in the parasitic wasp *Megaphragma* sp. (*Polilov, 2012*; *Polilov, 2017*).

Size range of Collembola is from 0.12 to 17 mm. Factors such as soil interstices space and humidity levels affect size reduction of collemblans. Thus, many collembolan genera include especially minute species (<500 µm), and therefore represent interesting models for research on miniaturization in arthropods. However, studies on the effects of miniaturization in collembolans have not been performed yet, and data on the anatomy of minute collembolans are extremely scarce. Previous studies on collembolan anatomy were based mostly on larger species (Table S1). Moreover, the majority of them were concentrated on specific systems only. *Lubbock (1873)* described the anatomy of several species, but studied only the largest muscles of the body, and the head musculature was not mentioned. *Fernald (1890)* examined the anatomy of *Anurida maritima*, but he studied only the muscles associated with the digestive system, and the excretory system was not mentioned in his study. *Willem (1900)* briefly described the anatomy of 12 species, but the muscular and excretory systems were not mentioned. *Prowazek (1900)* focused on the embryology and anatomy of both larvae and adults of *Isotoma grisea* and *Achorutes viaticus*, but the head musculature was not mentioned. *Denis (1928)* described the anatomy of *A. maritima*, *Onychiurus fimetarius*, and *Tomocerus catalanus*, but the reproductive system and musculature of the body (except the head musculature) were not mentioned. *Mukerji (1932)* described the digestive, nervous, and excretory systems, and partly the head musculature of *Protanura carpenteri*. In addition, there were several studies on the muscular system of *Orchesella cincta* (*Folsom, 1899*; *Bretfeld, 1963*), *Neanura muscorum* (*Bretfeld, 1963*), *Tomocerus longicornis* (*Lubbock, 1873*), *Tomocerus* spp. (*Eisenbeis & Wichard, 1975*), *Orchesella villosa*, *Isotomurus palustris*, *Podura aquatica*, and *Sminthurus viridis* (*Imms, 1939*), the digestive system of *Tomocerus flavescens* (*Humbert,*

*1979*), the digestive and excretory systems of *O. cincta* (*Verhoef et al., 1979*), *T. flavescens*, *A. maritima*, *N. muscorum*, *Friesea mirabilis*, *Brachystomella parvula*, *Odontella armata* (*Wolter, 1963*), and *Sminthurus fuscus* (*Willem & Sabbe, 1897*), the excretory system of *Onychiurus quadriocellatus* (*Altner, 1968*), *Tomocerus minor*, *Lepidocyrtus curvicollis* (*Humbert, 1975*), and *Orchesella rufescens* (*Philiptschenko, 1907*), the respiratory system of *S. viridis* (*Davies, 1927*), the nervous system of *Folsomia candida*, *Protaphorura armata*, and *Tetrodontophora bielanensis* (*Kollmann, Huetteroth & SchachPster, 2011*), the reproductive system of *Allacma fusca* (*Dallai et al., 2000*), *O. villosa* (*Dallai, Zizzari & Fanciulli, 2008*), *A. maritima* (*Lécaillon, 1902a*), and *Anurophorus laricis* (*Lécaillon, 1902b*). *Schaller (1970)* and *Hopkin (1997)* gave partial reviews of the above in their synthetic books on Collembola.

The genus *Mesaphorura* includes some of the smallest species of Collembola, the adults some of them are only 0.4 mm long (*Zimdars & Dunger, 1994*). The external morphology of *Mesaphorura* has been completely and thoroughly investigated for systematic and phylogenetic purposes (*Zimdars & Dunger, 1994*; *D'Haese, 2003*), but its internal morphology has never been described. The aim of this work is to study the anatomy of *Mesaphorura sylvatica* for the first time and to describe the effects of miniaturization in this species.

## MATERIALS AND METHODS

This complex anatomical study was conducted based on the methods for the study of microinsects described in previous papers (*Polilov, 2016a*; *Polilov, 2017*; *Polilov & Makarova, 2017*).

### Materials
Specimens of *Mesaphorura sylvatica* Rusek, 1971 were collected in September 2015 on a sand beach, on the bank of the Pirogovskoye Reservoir, Moscow Oblast, Russia, using the flotation method. The material was fixed in alcoholic Bouin's solution and stored in 70% ethanol.

### Scanning electron microscopy (SEM)
External morphology was studied using a Jeol JSM-6380 scanning electron microscope following critical point drying (Hitachi HCP-2) and sputter coating of samples with gold (Giko IB-3).

### Histology
The fixed material was dehydrated in a series of increasing ethanol solutions (in 70% and 95% for an hour, and twice in 100% for 30 min) and in acetone (twice in 100% for 30 min), and afterwards embedded in Araldite M (kept in a mixture of araldite and acetone 1:1 for a night, then twice in araldite for four hours, following polymerization at 60 °C for two days). The blocks were cut into series of cross section 1 μm thick and longitudinal section 0.5 μm thick using a Leica RM2255 microtome. These sections were stained with toluidine blue and pyronine.

### Three dimensional computer reconstruction (3D)

The sections were photographed using a Motic BA410 microscope with LED illumination source and ToupTek camera (5 MP). The resulting stack was then aligned and calibrated. 3D reconstructions were created in the program Bitplane Imaris 7.2 using the function of creating vector surface manually by outlining contours of structures on a series of slides. In addition, we processed the reconstructions with the functions of surface smoothing and rendering in the Autodesk Maya 2015 program.

### Measurements

The body length was measured using SEM images. Linear measurements were based on images of histological slides in the program Bitplane Imaris. Volumes of organs (Table S2) and of the whole body were calculated using the statistical module of Bitplane Imaris, as described in an earlier study (*Polilov & Makarova, 2017*). For all measurements we calculated means, n-number, and minimum and maximum values, where it was possible with a given sample size.

### Nomenclature

The names of morphological elements are based on *Folsom (1899)*, *Snodgrass (1935)*, *Bretfeld (1963)*, and *Bitsch (2012)*. The description of the musculature and abbreviations of muscles (Table S3) are based on *Folsom (1899)* for the head, *Bretfeld (1963)* for the thorax and abdomen, and *Eisenbeis & Wichard (1975)* for the ventral tube, with some additions. Muscles are named according to the nomenclature used for insects (*Friedrich & Beutel, 2008*; *Wipfler et al., 2011*). The following abbreviations are used in descriptions of muscles: O, origin; I, insertion.

## RESULTS

### General morphology

The body is from 313 to 492 μm ($M = 425$, $n = 4$) in length (Figs. 1A–1C), uniformly white in color. Jumping organ (furca), tenaculum and eyes are absent. It is important to note that the name "furca" is also applied to endoskeletal structures in thorax of hexapods ("furca-like structures" mentioned below). Most of the head is occupied by the brain, the suboesophageal ganglion, the mouthparts and the complex pseudotentorium; the prothorax is occupied by part of the suboesophageal ganglion, while the meso- and metathorax are occupied by the wide midgut and fat body; the abdomen is mainly occupied by the reproductive system, with the digestive system above it (Figs. 2A–2D, Fig. S1). All tagmata have well-developed musculature.

The body volume of *M. sylvatica* is about 0.79 nl.

### Skeleton

The cuticle thickness is 0.31–1.24 μm ($M = 0.57$, $n = 80$). The tergites are well-developed; the sclerites and pleurites are hardly distinguishable.

The inner skeletal structures are highly developed. A complex pseudotentorium (pst) (Figs. 3A–3B and Figs. 3D–3F) is situated in the head. Its body (bp) consists of a mandibular

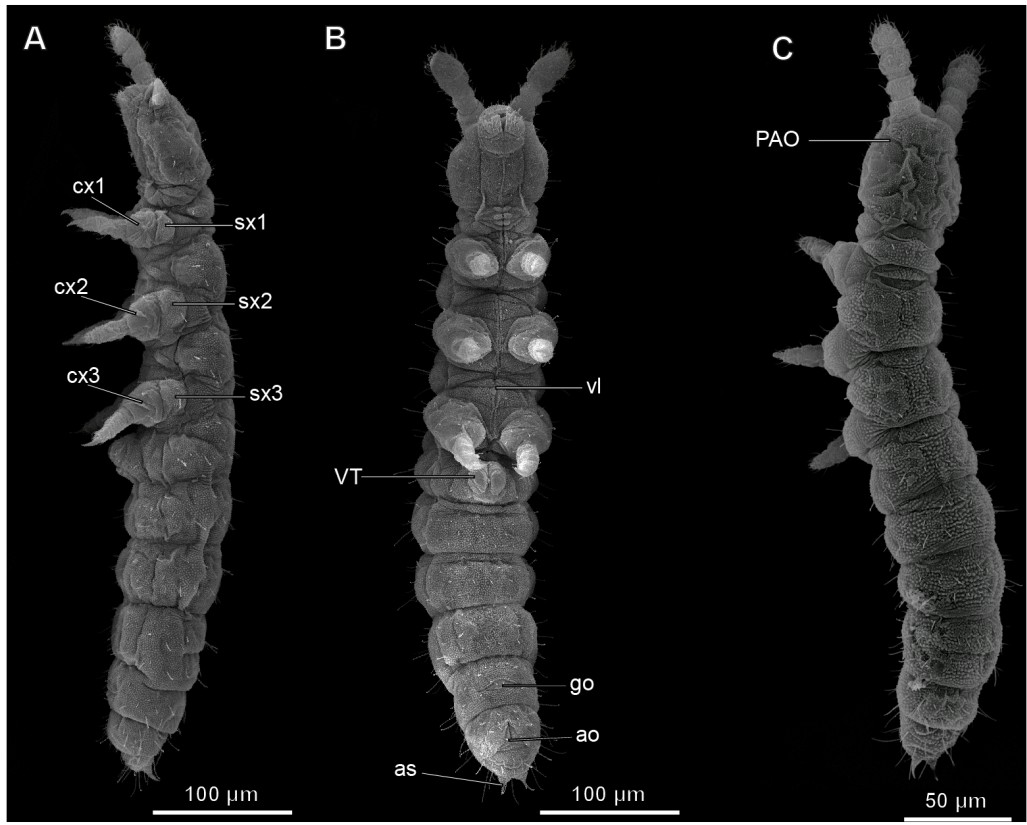

**Figure 1** **Habitus of *Mesaphorura sylvatica*, SEM.** (A) Lateral view; (B) ventral view; (C) dorsal view; ao, anal opening; as, anal spine; cx1, 2, 3—pro-, meso-, and metacoxae; go, genital opening; PAO, postantennal organ; sx1, 2, 3—pro-, meso, and metasubcoxae; vl, ventral line; VT, ventral tube. Furcula and eyes are absent.

tendon in the middle, which continues posteriorly into a thinner longitudinal endoskeletal connective. There is a pair of dorsal suspensory arms (dsa), connecting the structure with the head capsule anteriorly on the frons. The glossa is prolonged posteriorly into a pair of chitinous stalks (ful), called the posterior tentorial apodemes by *Koch (2000)*, or fulcra (*Denis, 1928*). They lie externally to the midline. The enlarged base of the stalk is called the foot (fo), and the foot underlies the cardo (car) of the maxilla (Mx). They seem to connect with the head capsule posteriorly possibly with some endoskeletal connectives. A pair of connecting arms (ca) (bras d'union, *Denis, 1928*) extend from the pseudotentorial plate downwards and are fused with the posterior tentorial apodemes. A pair of lateral arms (la) (bras latéraux, *Denis, 1928*) extend from the anterior part of the pseudotentorial plate, go upward and outward and are inserted into the head.

In *Orchesella cincta*, according to *Folsom (1899)*, there is also a chitinous rod, which is attached to the base of the lobe of the lacinia. The chitinous rod has a chitinous expansion (ce), which is the attachment site for several maxillary muscles. It is shown in our model as a part of the maxilla.

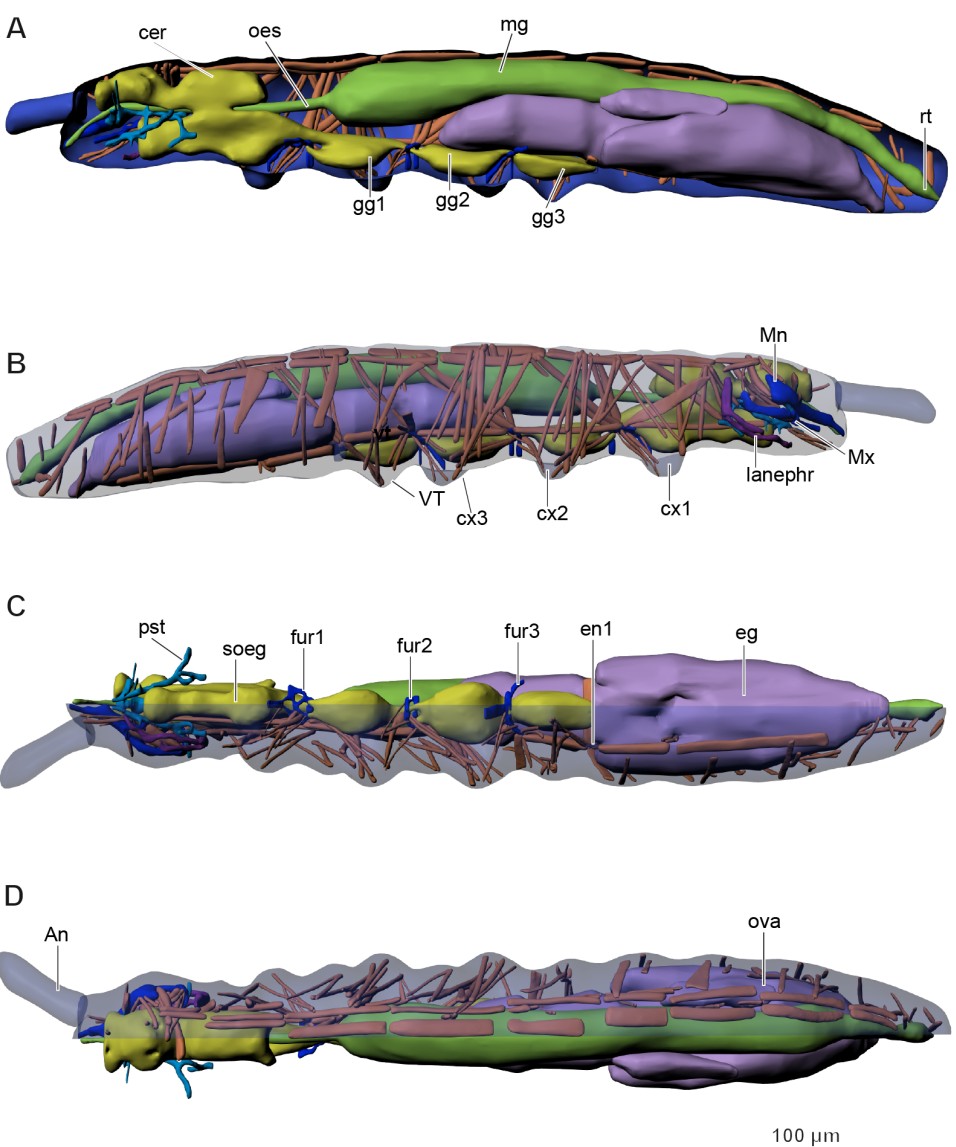

**Figure 2** **Internal morphology of *Mesaphorura sylvatica*, 3D.** Colors: blue, cuticle; light blue, tentorium; green, digestive system; yellow, central nervous system; brown, musculature; purple, reproductive system; dark violet, excretory system: (A) lateral internal view; (B) lateral external view; (C) ventral view; (D) dorsal view; An, antennae; cer, brain; cx1, 2, 3—pro-, meso-, and metacoxae; eg, ovary lobe with eggs; en1, endosternite; fur1, 2, 3—pro-, meso-, and metafurca-like structures; gg1, 2, 3+ag—pro-, meso-, and metathoracic+abdominal ganglia; lanephr, labial nephridia; mg, midgut; Mn, mandible; Mx, maxilla; oes, oesophagus; ova, ovary lobe without eggs; pst, pseudotentorium; rt, rectum; soeg, suboesophageal ganglion; VT, ventral tube. Paired structures (maxillae, mandible, labial nephridia, muscles) are shown on the right side only.

Antecostae are submarginal ridges near the anterior edges of the inner surface of the tergum with several body muscles attached to them.

Three ventral furca-like structures are branched and found in the thorax between the first and the second thoracic segments, between the second and the third thoracic segments, and

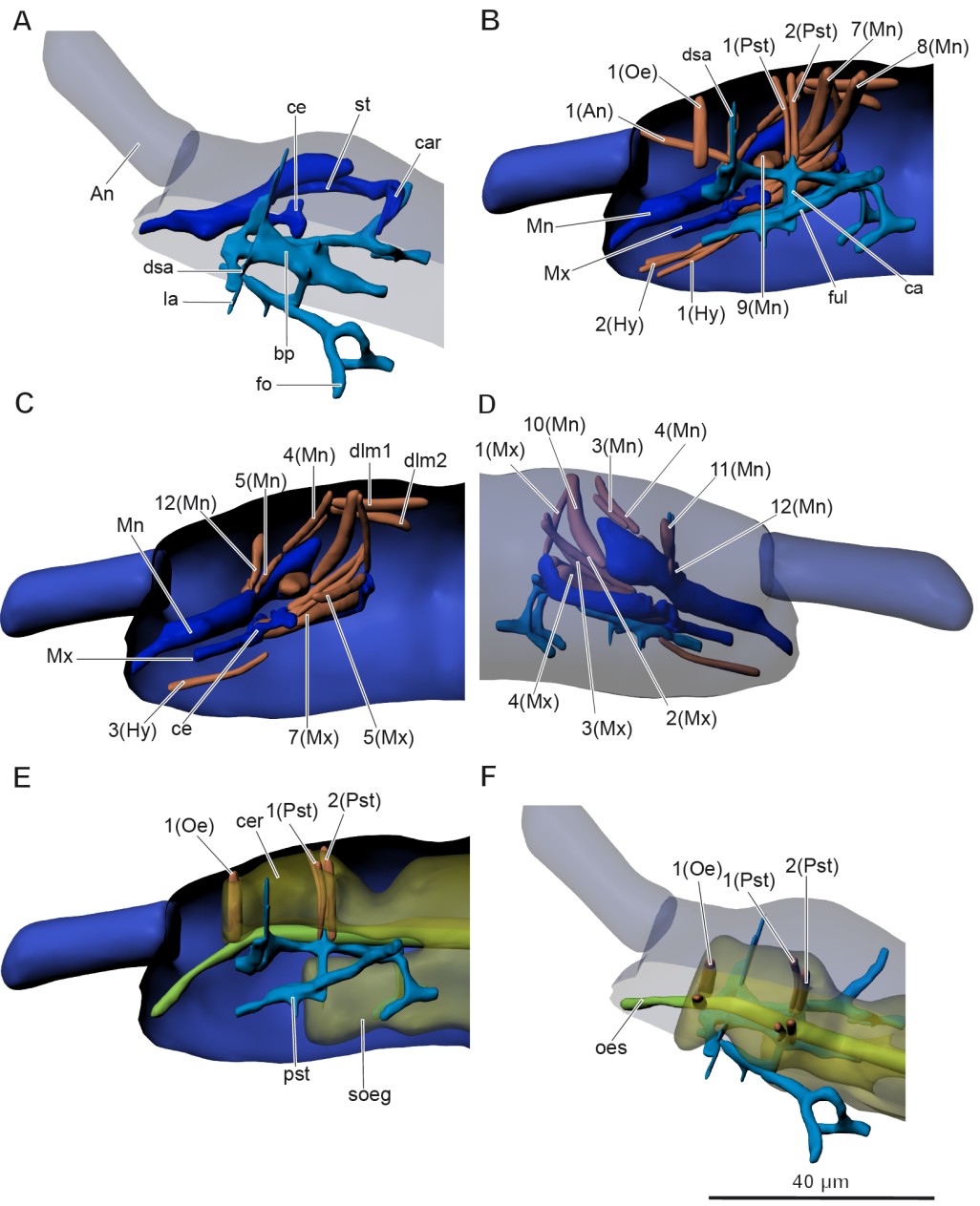

**Figure 3** **Anatomy of head in *Mesaphorura sylvatica*, 3D.** (A, F) Dorsolateral view, (B, C, E) lateral internal view, (D) lateral external view. An, antennae; bp, body of pseudotentorium; ca, connecting arm; car, cardo; ce, chitinous expansion; cer, cerebrum; dsa, dorsal suspensory arm; fo, foot; ful, fulcrum; la, lateral arm; Mn, mandible; Mx, maxillae; oes, oesophagus; pst, pseudotentorium; soeg, suboesophagal ganglion; st, stipes. Musculature see text. Paired structures (maxillae, mandible, muscles, except: 1(Oe), 1(Pst), 2(Pst) in E–F) are shown on the right side only.

between the third thoracic segment and the first abdominal segment (Fig. 2C). Additionally, there is a simple rectangular endosternite in the first abdominal segment.

The volume of the skeleton is about 0.048 nl (5.8% of the body volume).

### Digestive and excretory systems

The alimentary canal (Figs. 4A–4B) is shaped as a straight tube without loops or diverticula, extending from the anterior and ventral area of the head into the terminal abdominal segment. It is divided into fore-, mid-, and hindgut.

The slender foregut is circular in cross section and extends posteriorly from the oral cavity. It is divided into the pharynx and oesophagus (oes). The slender pharynx is about 4.2 μm in diameter ($M = 4.2$, $n = 8$). The oesophagus passes through the suboesophageal ganglion and leads into the thicker midgut (mg) at the level of the metathorax (around the fourth abdominal segment). The midgut consists of one layer of cells (6–8 cells in cross section). The oesophagus has one pair of muscles 1(Oe) (Figs. 3E and 3F). The first half of the midgut is circular in cross section, about 21.8 μm ($M = 24.3$, $n = 8$) in diameter. The second half is oval in cross section. The border between the midgut and the hindgut is indistinguishable. At around the sixth abdominal segment, the hindgut extends into the wider rectum (rt) with four pairs of muscles. The latter continues posteriad and terminates ventrally at the anus with three anal lobes in the last abdominal segment.

Labial nephridia (laneph), or tubular glands, the main excretory organs of collembolans, are found in the posterior half of the head (Figs. 4A–4B). Each nephridium is composed of a sac, a labyrinth, and a duct. The sac is situated posteriorly and continues anteriad into the labyrinth. The labyrinth follows a hardly distinguishable winding course and forms a loop. The labyrinth continues as the duct, which opens in the buccal cavity. Other head glands (anterior and posterior salivary glands, globular, or acinous glands, and antennal nephridia) were not found in *M. sylvatica*.

The volume of the digestive and excretory systems is about 0.68 nl (8.6% of the body volume).

### Nervous system

The nervous system (Figs. 4C–4D) consists of a supraoesophageal ganglion, or brain (cer), suboesophageal ganglion (soeg), and three thoracic ganglia. The brain extends from the bases of the antennae to the anterior part of the first thoracic segment. The brain fills the dorsal portion of the head, but narrows in the posterior portion of the head (beyond the beginning of the suboesophageal ganglion) and extends beyond the boundary between the head and the first thoracic segment. It terminates in the anterior half of the latter. The brain has a unique structure, with two pairs of apertures with one pair of oesophageal 1(Oe) and two pairs of pseudotentorial suspensory muscles 1(Pst), 2(Pst) running through them (Figs. 3E–3F). The suboesophageal ganglion (Figs. 3E and 4C) lies in the ventral portion of the head, starting at its middle, and continues to the distal margin of the first thoracic segment. Three large ventral thoracic ganglia shift their position by one segment: the first ganglion (gg1) lies in the mesothorax, the second one (gg2) is in the metathorax, and the third one (gg3) is in the first abdominal segment. They are interconnected by longitudinal

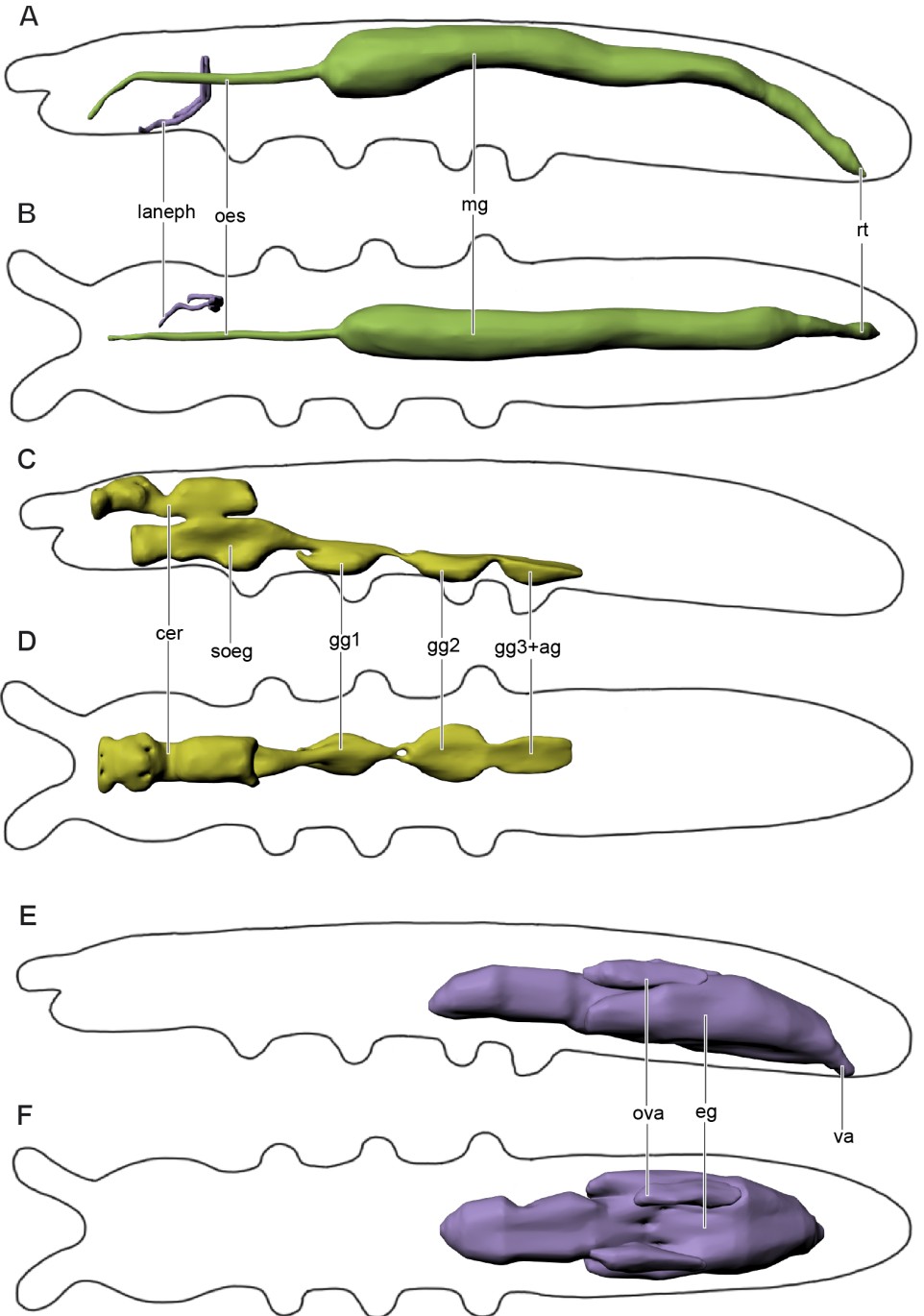

**Figure 4 Digestive and excretory (A, B), nervous (C, D), and reproductive (E, F) systems of *Mesaphorura sylvatica*, 3D.** (A, C, E) Lateral view; (B, D, F) dorsal view. cer, brain, eg—ovary lobe with eggs, gg1, 2, 3+ag—pro-, meso-, and metathoracic+abdominal ganglia; lanephr, labial nephridia; mg, midgut; oes, oesophagus; ova, ovary lobe without eggs; rt, rectum; soeg, suboesophageal ganglion; va, vagina. Paired structures (labial nephridia) are shown on the right side only.

cords in intersegments (one in each). As in all collembolans, the abdominal ganglia (ag) are fused with the third thoracic ganglion.

The volume of the central nervous system is about 0.051 nl (6.3% of body volume). The volume of the brain is about 0.016 nl (2.2% of the body volume).

## Muscular system

**Musculature of head** (Figs. 3B–3F, Table 1). *M. sylvatica* has 24 pairs of muscles in the head. Two of them are connected to the pseudotentorium, one to the antennae, nine to the mandibles, six to the maxillae, one to the oesophagus, and two of them are the dorsal lateral pairs of muscles. In addition, there are three very small obscure muscles that are connected to the hypopharynx. Two mandibular muscles 8(Mn) cross each other and attach to the opposite sides of the head.

**Musculature of thorax** (Figs. 5A–5C, Table 2).

Prothorax. *M. sylvatica* has 14 pairs of muscles in the prothorax. Two of them are dorsal longitudinal, one ventral longitudinal, two intersegmental dorsoventral, five dorsoventral, and four sterno-coxal.

Mesothorax. *M. sylvatica* has 19 pairs of muscles in the mesothorax. Compared to the prothorax, it has more dorsoventral (eight) and sterno-coxal (six) pairs of muscles.

Metathorax. *M. sylvatica* has 18 pairs of muscles in the mesothorax. Compared to the mesothorax, only one pair of muscles (III scm4) is absent.

**Musculature of abdomen** (Figs. 6A–6B, Table 3). *M. sylvatica* has 61 pairs of muscles and one unpaired muscle in the abdomen. Ten pairs of muscles are dorsal longitudinal, three ventral longitudinal, eight intersegmental dorsoventral, 29 dorsoventral. In addition, there are 11 pairs of muscles that are connected to the ventral tube. One unpaired transversal muscle connects abdominal endosternites of both sides.

The volume of the muscular system is about 0.038 nl (5.2% of the body volume).

## Reproductive system

The female reproductive system has been studied in detail (Figs. 4E–4F). The ovary is unpaired with three lobes. The largest lobe (eg) probably contains eggs, while two other, smaller lobes (ova) contain no eggs and lie dorsad of the largest one. The anterior portion of the ovary lies between the abdominal segments 2 and 3, while its posterior portion ends between abdominal segments 4 and 5. The oviduct is small, short and unpaired, leading to the vagina (va), the margins of which are indistinct. The vagina opens ventrally on abdominal segment 5 with a transverse reproductive orifice (gonopore).

The volume of the reproductive system is about 0.15 nl (18.9% of the body volume).

## Circulatory system and fat body

Organs of the circulatory system are absent, the system is represented by hemolymph in the body cavity. The fat body occupies all cavities between organs in the head, thorax, and abdomen. It consists of cells of various shape.

The volume of the circulatory system and fat body is about 0.44 nl (55.2% of the body volume).

**Table 1  Head muscle origins and insertions.**

| Abbrev. | Name | Origin | Insertion |
|---|---|---|---|
| 1(Pst) | M. craniotentorialis lateralis | Medial surface of frons, laterad of 2(Pst) | Dorsal surface of pseudotentorial plate, laterad of 2(Pst) |
| 2(Pst) | M. craniotentorialis medialis | Medial surface of frons, mediad of 1(Pst) | Dorsal surface of pseudotentorial plate, mediad of 1(Pst) |
| 1(An) | M. antennotentorialis | Lateral face of first antennal segment | Pseudotentorium |
| 3(Mn) | M. craniomandibularis posterior | Posterior surface of gena | Dorsolateral surface of basal ridge of mandible, posterad of 4(Mn) |
| 4(Mn) | M. craniomandibularis anterior | Frons, mediad of 3(Mn) | Dorsolateral surface of basal ridge of mandible, anteriad of 3(Mn), posteriad of 5(Mn) |
| 5(Mn) | M. tentoriomandibularis 1 | Anterior arm of pseudotentorium | Dorsolateral surface of mandible, anteriad of 4(Mn) |
| 7(Mn) | M. craniomandibularis 1 | Frons, along with 1(Mx), mediad of 10(Mn) | Ventroposterior area, outer angle of large triangular opening of mandible along with 8(Mn), 10(Mn) |
| 8(Mn) | M. craniomandibularis 2 | Posterior surface of frons, crossing median plane, posteriad of 7(Mn) | Ventroposterior area, outer angle of large triangular opening of mandible along with 7(Mn), 10(Mn) |
| 9(Mn) | M. tentoriomandibularis 2 | Base of pseudotentorium | Large triangular opening (median surface) of mandible |
| 10(Mn) | M. craniomandibularis 3 | Frons, laterad of 7(Mn) | Ventroposterior area, outer angle of large triangular opening of mandible along with 7(Mn), 8(Mn) |
| 11(Mn) | M. craniomandibularis 4 | Anterior surface of area antennalis, near antennal base, ventrad of 12(Mn) | Lateral surface of mandible, laterad of 12(Mn) |
| 12(Mn) | M. craniomandibularis 5 | Anterior surface of area antennalis, dorsad of 11(Mn) | Lateral surface of mandible, mediad of 11(Mn) |
| 1(Mx) | M. craniocardinalis | Dorsomedial area of occiput | Lateral edge of cardo, dorsad of 5(Mx) |
| 2(Mx) | M. craniostipitalis medialis | Posterior surface of gena, ventrad of 3(Mx) | Lateral edge of chitinous expansion, mediad of 3(Mx) |
| 3(Mx) | M. craniofurcalis lateralis | Posterior surface of gena, dorsad of 2(Mx) | Median edge of chitinous expansion, laterad of 2(Mx) |
| 4(Mx) | M. maxillaris internus 1 | Anterior surface of cardo | Dorsoventral surface of chitinous expansion |
| 5(Mx) | M. tentoriocardinalis | Base of pseudotentorium | Concavity of cardo, ventrad of 1(Mx) |
| 7(Mx) | M. maxillaris internus 2 | Median surface of stipes | Dorsolateral surface of chitinous expansion |
| 1–3(Hy) | M. craniohypopharyngealis | Anterioventral area of the head capsule | Hypopharynx |
| 1(Oe) | M. cranioesophagialis | Anteriomedial surface of area antennalis | Dorsal surface of oesophagus |
| dlm1 | M. occiputo-cranialis medialis | Occiput, mediad of dlm2 | Medial surface of frons, mediad of dlm2 |
| dlm2 | M. occiputo-cranialis lateralis | Occiput, laterad of dlm1 | Medial surface of frons, laterad of dlm1 |

## Respiratory system

Organs of the respiratory system (tracheae) are absent.

## DISCUSSION

We studied the anatomy of *M. sylvatica* to extend the knowledge on the anatomy of Collembola as well as to reveal possible miniaturization traits and to compare them to the

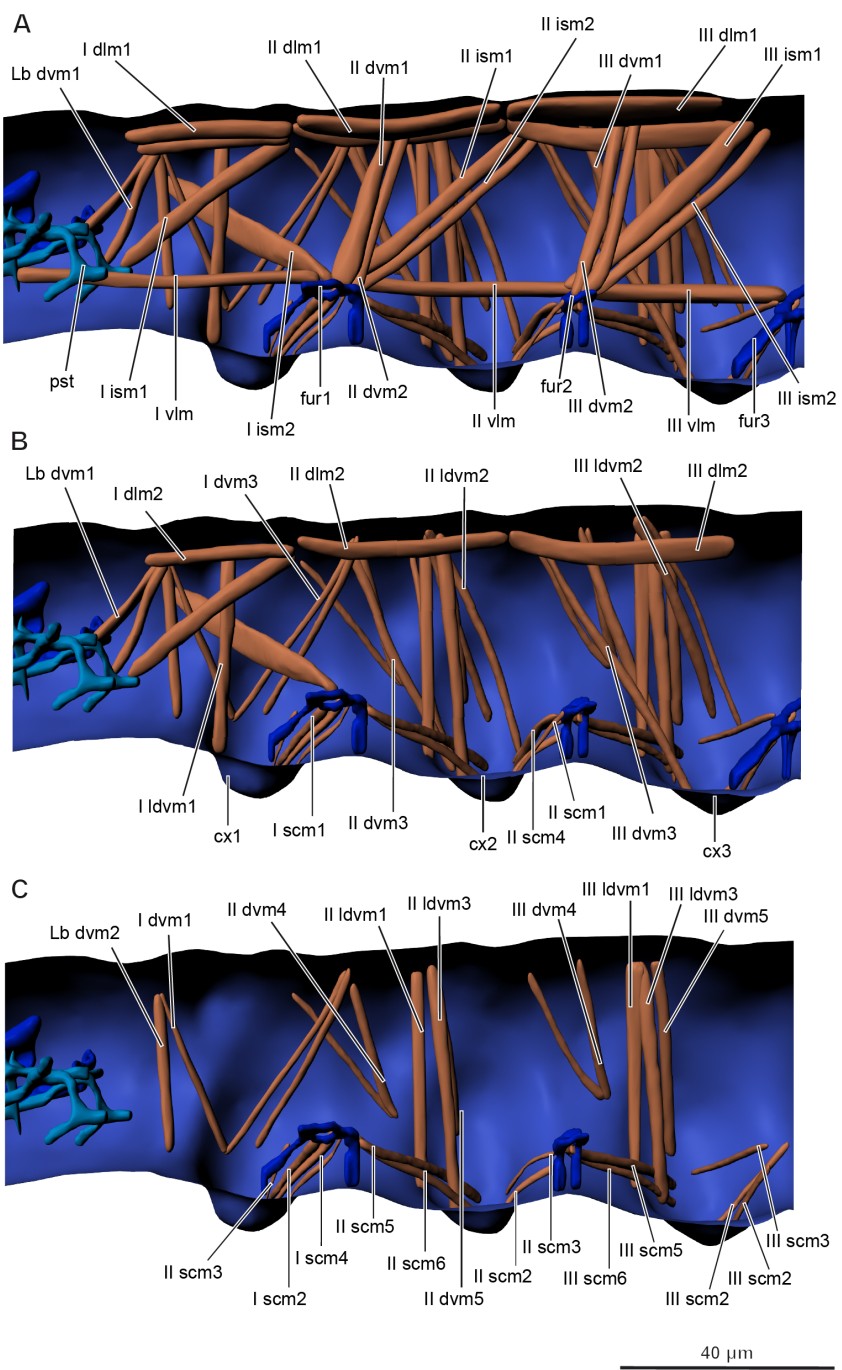

**Figure 5 Musculature of thorax in *Mesaphorura sylvatica*, 3D.** (A–C) Lateral internal view. cx1, 2, 3—pro-, meso-, and metacoxae, fur1, 2, 3—pro-, meso-, and metafurca-like structures; pst, pseudotentorium. Musculature see text.

**Table 2 Thoracic muscle origins and insertions.**

| Abbrev. | Name | Origin | Insertion |
|---|---|---|---|
| I dlm1 | M. antecosta-occipitalis medialis | Antecosta I, mediad of I dlm2 | Occiput, mediad of I dlm2 |
| I dlm2 | M. antecosta-occipitalis lateralis | Antecosta I, laterad of I dlm1 | Occiput, laterad of I dlm1 |
| I vlm | M. profurca-pseudotentoralis | Anterior part of profurca-like structure | Union arm of pseudotentorium |
| I ism1 | M. antecosta-pseudotentoralis | Antecosta I | Posterior area of fulcrum |
| I ism2 | M. profurca-occipitalis | Dorsal part of profurca-like structure | Dorsolateral area of occiput |
| I dvm1 | M. cervico-coxalis | Dorsolateral cervical membrane | Anterior procoxal rim |
| I dvm3 | M. pronoto-coxalis lateralis; two bands | Anterior region of pronotum | Lateral procoxal rim and anterior procoxal rim along with instertion of I dvm2 |
| I ldvm1 | M. pronoto-coxalis medialis | Anterolateral part of pronotum | Anterior procoxal rim |
| Lb dvm1 | M. occiputo-pseudotentoralis; two bands | Dorsal area of occipitale | Posterior area of fulcrum |
| Lb dvm2 | M. occiputo-cervicalis | Dorsal area of occipitale | Ventral cervical membrane |
| I scm1 | M. profurca-coxalis 1 | Anterior face of profurca-like structure | Posterior procoxal rim, laterad of I scm2 |
| I scm2 | M. profurca-coxalis 2 | Ventral face of profurca-like structure | Posterior procoxal rim, mediad of I scm1 |
| I scm3 | M. profurca-coxalis 3 | Ventral face of profurca-like structure along with I scm4 | Lateral procoxal rim |
| I scm4 | M. profurca-coxalis 4 | Ventral face of profurca-like structure along with I scm3 | Posteriolateral procoxal rim |
| II dlm1 | M. antecosta-antecostalis medialis | Antecosta II; mediad of II dlm2 | Antecosta III; mediad of II dlm2 |
| II dlm2 | M. antecosta-antecostalis lateralis | Antecosta II; laterad of II dlm1 | Antecosta III; laterad of II dlm1 |
| II vlm | M. profurca-mesofurcalis | Lateral part of profurca-like structure | Lateral part of mesofurca-like structure |
| II ism1 | M. profurca-antecostalis medialis | Lateral part of profurca-like structure; mediad of II ism2 | Antecosta III |
| II ism2 | M. profurca-antecostalis lateralis | Lateral part of profurca-like structure; laterad of II ism1 | Antecosta III |
| II dvm1 | M. mesonoto-profurcalis anterior | Dorsolateral part of profurca-like structure; anterior to II dvm2 | Mesonotum (middle of segment); anterior to II dvm2 |
| II dvm2 | M. mesonoto-profurcalis posterior | Dorsolateral part of profurca-like structure; posterior to II dvm1 | Mesonotum (middle of segment); posterior to II dvm1 |
| II dvm3 | M. mesonoto-coxalis; two bands | Lateral part of mesonotum (middle of segment) | Anterior face of mesocoxa |
| II dvm4 | M. mesonoto-subcoxalis anterior; two bands | Anterolateral part of mesonotum | Anterior border of mesosubcoxa |
| II dvm5 | M. metanoto-subcoxalis posterior | Posterolateral part of mesonotum | Ventral border of mesosubcoxa |
| II ldvm1 | M. mesonoto-coxalis anterior | Posterolateral part of mesonotum | Anterior mesocoxal rim |
| II ldvm2 | M. metanoto-subcoxalis; two bands | Posterolateral part of mesonotum | Posteroventral border of mesosubcoxa and posterior border of mesocoxal rim |
| II ldvm3 | M. metanoto-coxalis posterior | Posterolateral part of mesonotum | Anterior face of mesocoxa |
| II scm1 | M. mesofurca-coxalis 1 | Anterior face of mesofurca-like structure | Posterior mesocoxal rim, laterad of II scm2 |
| II scm2 | M. mesofurca-coxalis 2 | Ventrolateral face of mesofurca-like structure along with II scm4 | Posterior mesocoxal rim, mediad of II scm1 |

| Abbrev. | Name | Origin | Insertion |
|---|---|---|---|
| II scm3 | M. mesofurca-coxalis 3 | Anterior face of mesofurca-like structure | Lateral mesocoxal rim, laterad of II scm4 |
| II scm4 | M. mesofurca-coxalis 4 | Ventrolateral face of mesofurca-like structure along with II scm2 | Lateral mesocoxal rim, mediad of II scm3 |
| II scm5 | M. profurca-coxalis lateralis | Posterior face of profurca-like structure | Anteriolateral mesocoxal rim, laterad of II scm6 |
| II scm6 | M. profurca-coxalis medialis | Posterior face of profurca-like structure | Anteriolateral mesocoxal rim, mediad of II scm5 |
| III dlm1 | M. antecosta-antecostalis medialis | Antecosta ll, mediad of lll dlm2 | Antecosta lll, mediad of lll dlm2 |
| III dlm2 | M. antecosta-antecostalis lateralis | Antecosta ll, laterad of lll dlm1 | Antecosta lll, laterad of lll dlm1 |
| III vlm | M. mesofurca-metafurcalis | Posterolateral face of mesofurca-like structure | Anterolateral face of metafurca-like structure |
| III ism1 | M. antecosta-mesofurcalis anterior | Lateral face of mesofurca-like structure, anteriad of lll ism2 | Antecosta lll, anteriad of lll ism2 |
| III ism2 | M. antecosta-mesofurcalis posterior | Lateral face of mesofurca-like structure, posteriad of III ism1 | Antecosta lll, posteriad of lll ism1 |
| III dvm1 | M. metanoto-mesofurcalis anterior | Lateral face of mesofurca-like structure, anteriad of lll dvm2 | Lateral part metanotum (middle of segment), anteriad of lll dvm2 |
| III dvm2 | M. metanoto-mesofurcalis posterior | Lateral face of mesofurca-like structure, posteriad of lll dvm1 | Lateral part of metanotum (middle of segment), posteriad of lll dvm1 |
| III dvm3 | M. metanoto-coxalis; two bands | Anterolateral part of metanotum | Anterior metacoxal rim |
| III dvm4 | M. metanoto-subcoxalis anterior; two bands | Anterolateral part of metanotum | Anterior border of metasubcoxa |
| III dvm5 | M. metanoto-subcoxalis posterior | Posterolateral part of metanotum | Ventral border of metasubcoxa |
| III ldvm1 | M. metanoto-coxalis anterior | Lateral part of metanotum (middle of segment) | Anterior metacoxal rim |
| III ldvm3 | M. metanoto-coxalis posterior | Lateral part of metanotum (middle of segment) | Anterior face of metacoxa |
| III ldvm2 | M. metanoto-subcoxalis; two bands | Lateral part of metanotum (middle of segment) | Posteroventral border of metasubcoxa |
| III scm1 | M. metafurca-coxalis 1 | Ventromedial face of metafurca-like structure | Posterior metacoxal rim |
| III scm2 | M. metafurca-coxalis 2 | Ventrolateral face of metafurca-like structure | Lateral metacoxal rim |
| III scm3 | M. metafurca-coxalis 3 | Lateral face of metafurca-like structure | Lateral metacoxal rim |
| III scm5 | M. mesofurca-coxalis lateralis | Posterior face of mesofurca-like structure | Anteriolateral metacoxal rim, laterad of III scm6 |
| III scm6 | M. mesofurca-coxalis medialis | Posterior face of mesofurca-like structure | Anteriolateral metacoxal rim, mediad of III scm5 |

miniaturization effects discovered in microinsects and other minute arthropods. Moreover, we analyzed the relative volume of organs in *Mesaphorura sylvatica* in comparison with microinsects.

## Skeleton

The endoskeletal structures of *M. sylvatica* are well-developed as in larger species (Manton, 1964). The complex pseudotentorium has multiple arms, and the furca-like structures are branched. However, reductions seem to have affected the abdomen, in which we observed

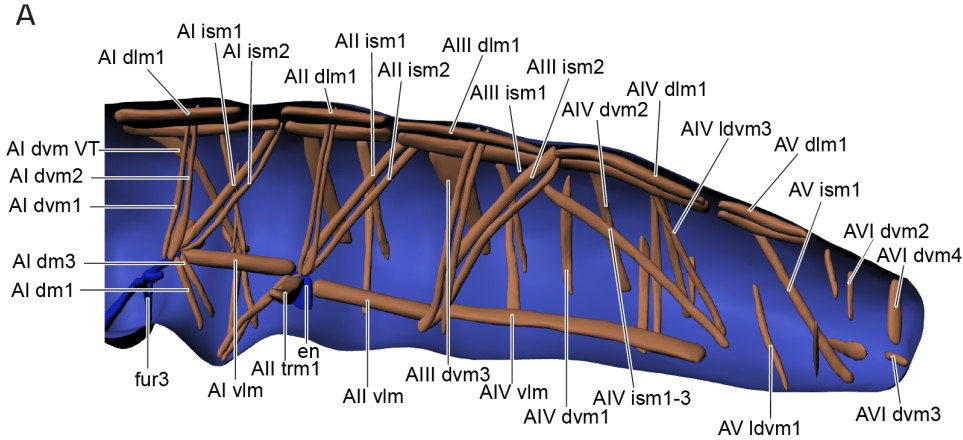

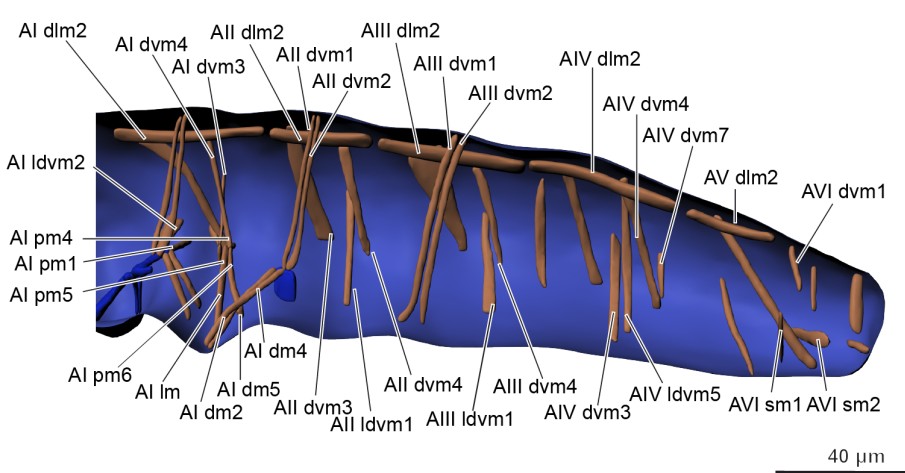

**Figure 6  Musculature of abdomen in *Mesaphorura sylvatica*, 3D.** (A–B) Lateral internal view. fur3, metafurca-like structure, en, endosternite. Musculature see text.

only one endosternite, in contrast to *Neanura muscorum*, in which five endosternites were found (*Bretfeld, 1963*). In microinsects, elements of the endoskeleton tend to fuse (*Polilov, 2015a*; *Polilov, 2015b*). Of all adult microinsects, only booklice have a complex tentorium (*Polilov, 2016b*), and all furcae are developed in thrips, beetles of the family Corylophidae, and wasps of the family Mymaridae (*Polilov & Beutel, 2010*; *Polilov & Shmakov, 2016*; *Polilov, 2016c*).

The relative volume of the skeleton of *M. sylvatica* is similar to the one of adult Paraneoptera of the same size, but notably smaller than the ones of other microinsects (both larvae and adults) of the same size (*Polilov & Makarova, 2017*). The smaller relative volume of the skeleton of *M. sylvatica*, compared to most microinsects, could be related to the differences in cuticle thickness (due to the fragmented epicuticle of all collembolans) and to the flightlessness of collembolans.

**Table 3  Abdominal muscle origins and insertions.**

| Abbrev. | Name | Origin | Insertion |
|---|---|---|---|
| AI dlm1 | M. antecosta-antecostalis medialis | Antecosta III, mediad of AI dlm2 | Antecosta IV, mediad of AI dlm2 |
| AI dlm2 | M. antecosta-antecostalis lateralis | Antecosta III, laterad of AI dlm1 | Antecosta IV, laterad of AI dlm1 |
| AI vlm | M. metafurca-endosternalis | Lateral face of metafurca-like structure | Endosternite |
| AI ism1 | M. antecosta-metafurcalis anterior | Antecosta IV, anteriad of AI ism2 | Lateral face of metafurca-like structure, anteriad of AI ism2 |
| AI ism2 | M. antecosta-metafurcalis posterior | Antecosta IV, posteriad of AI ism1 | Lateral face of metafurca-like structure, posteriad of AI ism1 |
| AI dvm1 | M. tergo-metafurcalis anterior | Middle region region of tergum, anteriad of AI dvm2 | Lateral face of metafurca-like structure, anteriad of AI dvm2 |
| AI dvm2 | M. tergo-metafurcalis posterior | Middle region region of tergum, posteriad of AI dvm1 | Lateral face of metafurca-like structure, posteriad of AI dvm1 |
| AI dvm3 | M. pleuro-pleuralis | Lateral wall of segment | Tendon system |
| AI dvm4 | M. tergo-pleuralis | Posterior region of tergum | Tendon system |
| AI ldvm1 | M. pleuro-metafurcalis | Lateral wall of segment | Lateral face of metafurca-like structure |
| AI dvm VT | M. tergo-pleuralis | Anterior region of tergum | Tendon system |
| AI lm | M. ventratubularis lateralis 1 | Base of ventral tube | Valva of ventral tube |
| AI pm1 | M. sterno-coxalis proximalis anterior 1 | Base of ventral tube | Metafurca-like structure |
| AI pm2 | M. sterno-coxalis proximalis anterior 2 | Base of ventral tube, anteriad of AI pm3, ventrad of AI pm4 | Tendon system |
| AI pm3 | M. sterno-coxalis proximalis anterior 3 | Base of ventral tube, posteriad of AI pm2, AI pm4 | Tendon system |
| AI pm4 | M. coxalis proximalis posterior 1 | Base of ventral tube, anteriad of AI pm3, dorsad of AI pm2 | Tendon system |
| AI dm1 | M. sterno-vesicularis | Anterior face of ventral tube, laterad of AI dm3 | Metafurca-like structure |
| AI dm2 | M. tergo-vesicularis anterior | Vesicles of ventral tube, ventrad of AI dm4 | Endosternite |
| AI dm3 | M. coxo-vesicularis | Anterior face of ventral tube, mediad of AI dm1 | Metafurca-like structure |
| AI dm4 | M. tergo-vesicularis posterior | Vesicles of ventral tube, laterad of AI dm2 | Endosternite |
| AI dm5 | M. coxo-vesicularis anterior | Posterior face of ventral tube | Tendon system |
| AII dlm1 | M. antecosta-antecostalis medialis | Antecosta IV, mediad of AII dlm2 | Antecosta V, mediad of AII dlm2 |
| AII dlm2 | M. antecosta-antecostalis lateralis | Antecosta lV, laterad of AII dlm1 | Antecosta V, laterad of AII dlm1 |
| AII vlm | M. endosterno-antecostalis | Endosternite | Antecosta V |
| AII ism1 | M. antecosta-endosternalis anterior | Antecosta V, anteriad of AII ism2 | Endosternite, anteriad of AII ism2 |
| AII ism2 | M. antecosta-endosternalis posterior | Antecosta V, posteriad of AII ism1 | Endosternite, posteriad of AII ism1 |
| AII dvm1 | M. tergo-endosternalis anterior | Middle region of tergum, anteriad of AII dvm2 | Endosternite, anteriad of AII dvm2 |
| AII dvm2 | M. tergo-endosternalis posterior | Middle region of tergum, posteriad of AII dvm1 | Endosternite, posteriad of AII dvm1 |
| AII ldvm1 | M. tergo-sternalis | Lateral wall of segment | Sternum |
| AII dvm3 | M. tergo-sternalis anterior | Anterior border of tergum, anteriad of AII dvm4 | Lateral border of sternum, anteriad of AII dvm4 |

**Table 3** (*continued*)

| Abbrev. | Name | Origin | Insertion |
|---|---|---|---|
| AII dvm4 | M. tergo-sternalis posterior | Lateral area of tergum, posteriad of AII dvm3 | Lateral border of sternum, posteriad of AII dvm3 |
| AII trm1 | M. endosterno-endosternalis | Inner surface of endosternite | Inner surface of endosternite (opposite side) |
| AIII dlm1 | M. antecosta-antecostalis medialis | Antecosta V, mediad of AIII dlm2 | Antecosta Vl, mediad of AIII dlm2 |
| AIII dlm2 | M. antecosta-antecostalis lateralis | Antecosta V, laterad of AIII dlm1 | Antecosta Vl, laterad of AIII dlm1 |
| AIII ism1 | M. antecosta-antecostalis medialis | Antecosta VI, mediad of AIII ism2 | Ventral area of antecosta V, mediad of AIII ism2 |
| AIII ism2 | M. antecosta-antecostalis lateralis | Antecosta VI, laterad of AIII ism1 | Ventral area of antecosta V, laterad of AIII ism1 |
| AIII dvm1 | M. tergo-antecostalis anterior | Middle region of tergum, anteriad of AIII dvm2 | Ventral area of antecosta V, anteriad of AIII dvm2 |
| AIII dvm2 | M. tergo-antecostalis posterior | Middle region of tergum, posteriad of AIII dvm1 | Ventral area of antecosta V, posteriad of AIII dvm1 |
| AIII dvm3 | M. tergo-sternalis anterior | Anterior border of tergum, anteriad of AIII dvm4 | Lateral border of sternum, anteriad of AIII dvm4 |
| AIII dvm4 | M. tergo-sternalis posterior | Lateral area of tergum, posteriad of AIII dvm3 | Lateral border of sternum, posteriad of AIII dvm3 |
| AIII ldvm1 | M. tergo-sternalis | Lateral wall of segment | Sternum |
| AIV dlm1 | M. antecosta-antecostalis medialis | Antecosta Vl, mediad of AlV dlm2 | Antecosta VII, mediad of AlV dlm2 |
| AIV dlm2 | M. antecosta-antecostalis lateralis | Antecosta Vl, laterad of AlV dlm1 | Antecosta VII, laterad of AlV dlm1 |
| AlV vlm | M. antecosta-antecostalis | Antecosta V | Antecosta VII |
| AIV ism1 | M. antecosta-antecostalis | Dorsal part of antecosta VI | Ventral part of antecosta VII |
| AIV dvm1 | M. tergo-sternalis posterior | Anterior border of tergum, posteriad of AIV dvm3 | Posterior border of sternum, posteriad of AIV dvm3 |
| AIV dvm2 | M. tergo-sternalis 1 | Lateral wall of segment | Sternum, along with AIV ldvm5 |
| AIV dvm3 | M. tergo-sternalis anterior | Anterior border of tergum, anteriad of AIV dvm1 | Lateral board of sternum, anteriad of AIV dvm1 |
| AIV ldvm3 | M. tergo-antecostalis | Posterior region of tergum | Ventral part of antecosta VII |
| AIV ldvm4 | M. tergo-sternalis 2 | Posterior region of tergum | Lateral board of sternum |
| AIV ldvm5 | M. tergo-sternalis 3 | Posterior region of tergum | Sternum, along with AIV dvm2 |
| AIV ldvm7 | M. tergo-sternalis 4 | Lateral wall of segment | Lateral board of sternum |
| AV dlm1 | M. antecosta-antecostalis medialis | Antecosta VII, mediad of AV dlm2 | Antecosta VIII, mediad of AV dlm2 |
| AV dlm2 | M. antecosta-antecostalis lateralis | Antecosta VII, laterad of AV dlm1 | Antecosta VIII, laterad of AV dlm1 |
| AV ism1 | M. tergo-intersegmentalis | Anterior region of tergum | Intersegmental area between 5th and 6th segments |
| AV ldvm1 | M. tergo-sternalis | Lateral wall of segment | Sternum |
| AVl sm1 | M. sterno-rectalis 1 | Posteriolateral board of sternum | Rectum, posteriad of AVI sm1 |
| AVI sm2 | M. sterno-rectalis 2 | Lateral board of sternum | Rectum, anteriad of AVI sm2 |
| AVl dvm1 | M. tergo-rectalis | Anterior region of tergum | Rectum, anteriad of AVI dvm2 |
| AVI dvm2 | M. tergo-sternalis 1 | Central region of tergum | Rectum, posteriad of AVI dvm1 |
| AVI dvm3 | M. tergo-sternalis 2 | Posterior region of tergum | Dorsal anal lobe |
| AVl dvm4 | M. tergo-sternalis 3 | Lateral wall of segment | Lateral anal lobe |

## Nervous system

Unlike those of larger species of Collembola (*Lubbock, 1873*; *Kollmann, Huetteroth & SchachPster, 2011*), the brain and suboesophageal ganglion of *M. sylvatica* extend into the prothorax. The three thoracic ganglia of *M. sylvatica* shift their position by one segment posteriorly. In larger collembolan species, an extension of the metathoracic ganglion to the first abdominal segment was reported (*Lubbock, 1873*; *Hopkin, 1997*). The brain and the suboesophagal ganglion are situated close to each other; they are connected in the neck region. The central nervous system is symmetrical and displays moderate concentration and oligomerization of ganglia. Similar degrees of concentration and oligomerization of the central nervous system are found in adult booklice of the family Liposcelididae (*Polilov, 2016b*) and in adult thrips (*Polilov & Shmakov, 2016*).

The shift of different parts of the brain into the prothorax has been described in thrips larvae (*Polilov & Shmakov, 2016*), adults and larvae of Ptiliidae (*Polilov & Beutel, 2009*), adults and larvae of Corylophidae (*Polilov & Beutel, 2010*), larvae of Scydmaenidae (*Jałoszyński, Hünefeld & Beutel, 2012*), larvae of Hydroscaphidae (*Beutel & Haas, 1998*), adults of Sphaeriusidae (*Yavorskaya et al., 2018*), and larvae of Strepsiptera (*Beutel, Pohl & Hunefeld, 2005*).

The nervous system of *M. sylvatica* shows unique changes in the brain with two pairs of apertures and three pairs of muscles running through them. This feature has not been described in studies of the nervous system of larger collembolans (*Lubbock, 1873*; *Hopkin, 1997*; *Kollmann, Huetteroth & SchachPster, 2011*) or microinsects (*Polilov, 2015a*; *Polilov, 2015b*).

The relative volume of the central nervous system of *M. sylvatica* is similar to the one of tiny adult Coleoptera, smaller than those of minute adult Hymenoptera and Paraneoptera larvae, and greater than those of adult Paraneoptera of the same size (*Polilov & Makarova, 2017*). Such small relative volume can be explained by the tendency of the nervous system of microinsects to increase as the body size decreases (*Polilov & Makarova, 2017*). It is also supported by the fact that representatives of adult minute Paraneoptera have greater body size, but smaller relative volume of the central nervous system. The smaller relative volume of the central nervous system of *M. sylvatica* compared to the ones of minute adult Hymenoptera and Paraneoptera larvae could be related to better pronounced effects of miniaturization in adult Hymenoptera and Paraneoptera larvae of the same size.

The relative volume of the brain of *M. sylvatica* is similar to the one of adult Coleoptera of the same size, slightly greater than the one of minute adult Paraneoptera, and notably smaller than the ones of other microinsects of the same size (*Polilov & Makarova, 2017*). The smaller relative volume than in most microinsects of the same size could possibly be related to the absence of flight organs and eyes (*Jordana, Baquero & Montuenga, 2000*).

## Circulatory system and fat body

The circulatory system of *M. sylvatica* is simplified, heart or vessels are absent. Most of the body cavities of *M. sylvatica* are filled with the fat body. In larger collembolan species, there is a heart with 2–6 pairs of ostia and an aorta (*Fernald, 1890*; *Denis, 1928*; *Imms, 1957*; *Schaller, 1970*). The same reduction as in *M. sylvatica* was observed in adults and

larvae of beetles of the family Ptiliidae (*Polilov, 2005*; *Polilov & Beutel, 2009*), larvae of booklice of the family Liposcelididae (*Polilov, 2016b*), adult hymenopterans of the family Trichogrammatidae (*Polilov, 2016d*; *Polilov, 2017*), tardigrades (*Gross et al., 2019*), and some chelicerates (*Dunlop, 2019*). In microinsects, it is assumed that the diffusion of metabolites is sufficient enough for the transport between the organs (*Polilov, 2008*; *Polilov & Beutel, 2009*), which is, apparently, also the case of *M. sylvatica.*

The relative volume of the circulatory system and fat body of *M. sylvatica* is particularly great, greater than the ones of other microinsects of the same size except Paraneoptera and Coleoptera larvae (*Polilov & Makarova, 2017*). The high relative volume could be related to the importance of the fat body in excretion in Collembola, where excretory products are stored (*Schaller, 1970*).

## Female reproductive system

The female reproductive system of *M. sylvatica* consists of unpaired ovaries and oviducts. In larger species, it consists of paired ovaries, oviducts and accessory glands, and an unpaired spermatheca. We did not observe accessory glands and spermatheca in *M. sylvatica,* which may be explained by the fact that they are hardly recognizable even in larger species (*Schaller, 1970*; *Dallai, Zizzari & Fanciulli, 2008*). The same changes were observed in beetles of the family Ptiliidae, in which both sexes have unpaired structures (*Polilov & Beutel, 2009*).

The relative volume of the reproductive system of *M. sylvatica* is also particularly great; it is smaller only than the ones of some minute adult Coleoptera and minute Hymenoptera (*Polilov & Makarova, 2017*). The greater relative volume compared to those of most microinsects of the same size could be related to the relative egg size increase with decreasing body size (*Polilov, 2016a*).

## Digestive and excretory systems

The digestive system of *M. sylvatica* is least modified, compared to larger species (*Lubbock, 1873*; *Folsom, 1899*; *Wolter, 1963*; *Schaller, 1970*). It is straight, without loops or diverticula. Among microinsects, only larvae of booklice of the family Liposcelididae have no loops or pronounced bends (*Polilov, 2016b*). No salivary glands are found in *M. sylvatica*. Salivary glands in some microinsects are absent as a result of miniaturization (*Polilov, 2015a*; *Polilov, 2015b*). We did not observe any muscles of the midgut, an absence of which is also a common trait among minute insects (*Polilov, 2016a*).

*M. sylvatica* has a pair of labial nephridia, but lacks other head glands (acinous glands and antennal nephridia) that are present in larger collembolan species (*Wolter, 1963*). In microinsects, Malpighian tubes do not disappear, but their number decreases (*Polilov, 2015a*).

The relative volume of the digestive and excretory systems of *M. sylvatica* is similar to those of most minute Coleoptera, notably smaller than those of minute Paraneoptera, but greater than the ones of Hymenoptera and some Coleoptera species of the same size (*Polilov & Makarova, 2017*).

## Musculature

The muscular system of *M. sylvatica* is reduced in number, compared to those of larger collembolan species. *M. sylvatica* has 24 pairs of muscles in the head, 51 in the thorax, and 61 in the abdomen (and 1 unpaired muscle); 136 pairs in total. The total number of muscles of all tagmata have not been studied in any single species of springtails, which limits the possible comparison of our results with previous studies.

It is difficult to compare head muscles in Collembola. *Folsom (1899)* described at least 47 pairs of muscles in the head of a large collembolan *Orchesella cincta* associated with the digestive system and mouthparts. There are 26 pairs of muscles associated with mouthparts (labrum, labium, maxilla, mandible). In addition, he noted 20 pairs of muscles associated with both pharynx and oesophagus (seven of them are the ventral dilators of the pharynx, which he later classified as tentorial muscles), but he did not designate them. He mentioned the presence of tentorial muscles (dilators of pharynx, antennal, and muscles connected to the head), but did not specify their number. *Folsom (1899)* also mentioned two muscles of the palpi, but it is not clear whether he meant two pairs of muscles or two muscles in total. *Denis (1928)* described at least 73 pairs of muscles in the head of a large collembolan *Anurida maritima* associated with mouthparts, pseudotentorium and the digestive system. He divided the muscles of the head into several groups, but he specified the number of muscles only for some of them. For the mouthparts (maxilla and mandible), he remarked that for some muscles he drew a single bundle that included several muscles, but he did not specify their number. There are at least 17 pairs of muscles associated with the maxilla and mandible in *A. maritima*. Moreover, he described all tentorial muscles, and there are at least 45 of them (14 of those are the ventral dilators of pharynx). In addition, he mentioned superlingual muscles (the number was not given), suspensors of the atrium (three pairs), eight pairs of antennal muscles (excluding the muscles inside the antenna), and five muscles associated with the epipharynx and pharynx. He did not specify the number of muscles of the labium. Moreover, he described the groups of muscles of the heads of two larger species *Onychiurus fimetarius* and *Tomocerus catalanus*, but he compared them to *A. maritima*, without providing details on exact numbers. With *T. catalanus* he referred to the study of *Hoffmann (1908)*, who described at least 53 pairs of muscles in the head of another large collembolan *Tomocerus plumbeus*. There are 35 muscles associated with mouthparts (maxilla, mandible, and labium), 15 with the pharynx, and three with the glossa. *Hoffmann (1908)* also mentioned the presence of tentorial muscles, but did not specify their number. In *M. sylvatica*, there are 15 pairs of muscles associated with maxilla and mandible, one pair of antennal muscles, one pair of oesophageal muscles, two pairs of suspensory pseudotentorial muscles, and three pairs of muscles, possibly associated with the hypopharynx. Moreover, there are two pairs of dorsal longitudinal muscles, while this group of muscles was mentioned, but not described in the literature. In this study, we have not described any other tentorial muscles, except those mentioned above, and internal antennal muscles due to their small size. *M. sylvatica* has 15 pairs of muscles of maxilla and mandible, which is less than in larger collembolans such as *O. cincta* (20), *A. maritima* (17), and *T. plumbeus* (29). *M. sylvatica* does not have muscles of labium or labrum, while there are six pairs of these in *O. cincta* and at least six pairs in *T. plumbeus*. *M. sylvatica* has

only one pair of dorsal dilators of the oesophagus and no dorsal dilators of the pharynx, which is fewer than in larger collembolans *O. cincta* (13), *T. plumbeus* (15).

A total of 51 pairs of muscles were described in the thorax of *Neanura muscorum* (*Bretfeld, 1963*) and a total of 37 pairs of muscles were described in the thorax of *O. cincta* (*Bretfeld, 1963*). Muscles associated with legs have not been described in that study. We found 36 pairs of muscles not associated with the legs in the thorax of *M. sylvatica* and 17 pairs of muscles associated with legs. There is a greater similarity between the muscles of *M. sylvatica* and *O. cincta* (both species have greater numbers of dorsoventral muscles) than between the muscles of *M. sylvatica* and *N. muscorum*. *M. sylvatica* lacks several dorsoventral and intersegmental muscles, while the amount of longitudinal muscles remain the same. It is important to note that *Bretfeld (1963)* described several muscles in the thorax as muscles possibly associated with the head. We describe two of them, Lb dlm1 and Lb dlm2, as dlm1 and dlm2 in the section on the muscles of the head. A total of 52 pairs and nine unpaired muscles were described in the abdomen of *N. muscorum* (*Bretfeld, 1963*) and a total of 45 pairs of muscles were described in the abdomen of *O. cincta* (*Bretfeld, 1963*). No muscles associated with the ventral tube, rectum, or anal lobes were described in these collembolans, except one in one species (VTm in *O. cincta*). We found 43 pairs and one unpaired muscle in the abdomen of *M. sylvatica*, not connected to the ventral tube, as well as 11 pairs of muscles associated with the ventral tube, four with the rectum, and two with the anal lobes. As for the thorax, *M. sylvatica* lacks many dorsoventral muscles, some intersegmental muscles and almost all transverse unpaired muscles.

Minute adult Coleoptera have 19 or 20 pairs of muscles in the head (*Sericoderus lateralis* and *Mikado* sp., respectively) and 48 or 49 pairs of muscles in the thorax (*Mikado* sp. and *S. lateralis*, respectively) (*Polilov & Beutel, 2009*; *Polilov & Beutel, 2010*). Compared to them, the number of the head pairs of muscles (24) and thoracic pairs of muscles (51) in *M. sylvatica* is slightly greater. Larvae of minute Coleoptera have 16 pairs of muscles in the head (*Mikado* sp. and *S. lateralis*) (*Polilov & Beutel, 2009*; *Polilov & Beutel, 2010*) and 46 (*Mikado* sp., first instar larva), 52 (*Mikado* sp., last instar larva) 63 (*S. lateralis*, first instar larva) or 64 (*S. lateralis*, last instar larva) pairs of muscles in the thorax. The number of thoracic pairs of muscles in the minute *M. sylvatica* (51) is close to the number of thoracic muscles in the last instar larvae of *Mikado* sp., but smaller than in larvae of *S. lateralis*. Minute Hymenoptera have 18 (*Megaphragma mymaripenne*, *Trichogramma evanescens*) (*Polilov, 2016d*; *Polilov, 2017*), or 20 (*Anaphes flavipes*) (*Polilov, 2016c*) muscles in the head and 45 (*M. mymaripenne*), 50 (*A. flavipes*), 52 (*T. evanescens*), or 53 (*Gonatocerus morrilli*) (*Vilhelmsen, Mikó & Krogmann, 2010*) muscles in the thorax. Compared to them, the number of the head muscles of *M. sylvatica* (24) is slightly greater, but the number of the thoracic muscles of this species (51) is greater only compared to those of *M. mymaripenne* and *A. flavipes*. Minute adult booklice *Liposcelis bostrychophila* have 33 pairs of muscles in the head and 57 pairs of muscles in the thorax (*Polilov, 2016b*). The larvae of *L. bostrychophila* have 29 pairs of muscles in the head and 55 pairs of muscles in the thorax (*Polilov, 2016b*). Compared to both larvae and adults, the number of the head pairs of muscles (24) and thoracic pairs of muscles (51) of *M. sylvatica* is notably smaller. Minute adult thrips *Heliothrips haemorrhoidalis* have 19 pairs of

muscles in the head and 60 pairs of muscles in the thorax (*Polilov & Shmakov, 2016*). Compared to them, the number of the head pairs of muscles (24) of *M. sylvatica* is greater, but the number of the thoracic pairs of muscles (51) is notably smaller. Larvae of *H. haemorrhoidalis* have 18 pairs of muscles in the head and 41 pairs of muscles in the thorax (*Polilov & Shmakov, 2016*). Compared to them, the number of the head pairs of muscles (24) and thoracic pairs of muscles (51) of *M. sylvatica* is notably greater. Minute Neuroptera *Coniopteryx pygmaea* (*Randolf, Zimmermann & Aspöck, 2017*) have 46 pairs of muscles in the head. Compared to them, the number of the head pairs of muscles (24) of *M. sylvatica* is notably smaller.

In all studied microinsects, there are three groups of abdominal muscles: dorsoventral, dorsal longitudinal, and ventral longitudinal (*Polilov, 2016a*). All three groups are present in the abdomen of *M. sylvatica*.

To sum up, the musculature system of *M. sylvatica* shows minor reductions in the number of muscles compared to larger collembolan species. In the head, absent muscles such as some mandibular retractors or maxillary adductors are not unique, and other muscles, present in *M. sylvatica*, have the same function (*Folsom, 1899*). In thorax and abdomen, ventral and dorsal longitudinal, and intersegmental muscles are present in full amount in *M. sylvatica*, but many dorsoventral muscles are absent. Nevertheless, they most likely do not differ in function from the dorsoventral muscles, present in *M. sylvatica*. The reduction in total number of muscles in *M. sylvatica* does not seem to affect any abilities of *M. sylvatica* to move. Studies on microinsects also show that the changes in musculature are minor, and this system is rather conserved (*Polilov, 2015a*). The number of muscles in *M. sylvatica* is slightly greater than those in most microinsects.

The relative volume of the musculature of *M. sylvatica* is smaller than those of other microinsects of the same size except Coleoptera larvae (*Polilov & Makarova, 2017*). The smaller relative volume compared to those of other microinsects could be explained by the absence of flight musculature.

## CONCLUSIONS

We have studied the anatomy of the minute collembolan *M. sylvatica* for the first time. We show that, despite the small body size, some systems (the highly developed elements of the endoskeleton, or the complex musculature system) are not greatly changed compared to larger relatives.

We revealed possible miniaturization effects; most of them are found in microinsects too (the absence of organs of the circulatory system, unpaired ovaries and oviducts of the female reproductive system, absence of midgut musculature and salivary glands, reduction of some muscles).

Finally, we found some unique features in the anatomy of *M. sylvatica*: two pairs of apertures in the brain with three pairs of muscles going through it.

Reduction in size leads to changes in different organs and organ systems, giving us perspective on physical constrains of size limit. Studying miniaturization can also bring us further understanding on successful diversification of animals. Collembola is a highly

diversified group of terrestrial arthropods with many extremely reduced in size species. Therefore, it is crucial to study anatomical changes in other minute collembolans to broaden our knowledge of miniaturization in animals.

## ACKNOWLEDGEMENTS

We thank Natalia Kuznetsova (Moscow State Pedagogical University) and Pyotr Petrov (Moscow State University) for their helpful discussions.

### Funding

The study was supported by the Russian Science Foundation (project no. 19-14-00045). The funders had no role in study design, data collection and analysis, decision to publish, or preparation of the manuscript.

### Grant Disclosures

The following grant information was disclosed by the authors:
Russian Science Foundation: 19-14-00045.

### Competing Interests

The authors declare there are no competing interests.

### Author Contributions

- Irina V. Panina performed the experiments, analyzed the data, prepared figures and/or tables, authored or reviewed drafts of the paper, approved the final draft.
- Mikhail B. Potapov authored or reviewed drafts of the paper, approved the final draft, collected and identified specimens.
- Alexey A. Polilov conceived and designed the experiments, performed the experiments, authored or reviewed drafts of the paper, approved the final draft.

### Data Availability

The morphological study 3D-reconstruction is available at Zenodo: Panina Irina, Potapov Mikhail, & Polilov Alexey. (2019, June 18). Anatomy of collembola Mesaphorura sylvatica (Hexapoda: Collembola: Tullbergiidae). Zenodo. http://doi.org/10.5281/zenodo.3249103

### Supplemental Information

Supplemental information for this article can be found online at http://dx.doi.org/10.7717/peerj.8037#supplemental-information.

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
