# Peer review of "Effects of miniaturization in the anatomy of the minute springtail Mesaphorura sylvatica (Hexapoda: Collembola: Tullbergiidae)"

_PeerJ, doi:10.7717/peerj.8037_

## Round 0.1 · original submission · Major Revisions

Dear Dr. Panina and colleagues:

Thanks for submitting your manuscript to PeerJ. I have now received five independent reviews of your work, and as you will see, the reviewers raised some concerns about the review. Despite this, these reviewers are optimistic about your manuscript and the potential impact it will have on research communities studying springtails. Thus, I encourage you to revise your manuscript, accordingly, taking into account all of the concerns raised by both reviewers.

Please note that four of the five reviewers provided marked-up copies of your manuscript. Please address these concerns as well as the general comments by the reviewers.

While the concerns of the reviewers are relatively minor, this is a major revision to ensure that the original reviewers have a chance to evaluate your responses to their concerns.

I look forward to seeing your revision, and thanks again for submitting your work to PeerJ.

Good luck with your revision,

-joe

·

Basic reporting

This is an accurate and welcome contribution to the knowledge of morphology in miniaturized arthropods, the first ever for a member of the Collembola. The ms. reveals both the technical skills and the excellent familiarity with the subject of miniaturization, as expected from the AA.
Besides a few truly minor points annotated on the returned copy of the ms., I have only a serious point to raise, one however that can be addressed easily and quickly, that is the very inadequate figure legends, which are largely incomplete or inaccurate. To give one example, the legend for Figure 2 does not give an explanation for lanephr, Mn, Mx, pst, fur1,2,3, en1, eg, ova, and lists 'an' rather than 'AN' as in the figure.

Experimental design

no comment

Validity of the findings

no comment

Reviewer 2 ·

Basic reporting

The article is generally well written and clear, with a proper use of the English language. It represent a coherent body of work that constitute an appropriate unity of publication. I recommend that the authors spend some more words in the introduction and in the conclusion to place the topic in a broader context, in particular by explaining how knowledge about miniaturization can improve our understanding of the biology of animals and possibly by providing some examples of it. The data to support the article is clear (figures) though I suggest that some long stretches of text can be avoided by summarizing results and observations in other animals in one (or multiple) tables.

Experimental design

No comment

Validity of the findings

no comment

Additional comments

The paper is well written and the content is sound and scientifically valid. I think that the flow can be improved in a couple of parts to improve readibility, possibly by summarizing information, maybe in the form of a table (see my comment in the pdf).

Annotated reviews are not available for download in order to protect the identity of reviewers who chose to remain anonymous.

Reviewer 3 ·

Basic reporting

This paper has potential for making a valuable contribution toward our understanding of the evolution of extremely small body sizes in metazoan animals. It is mainly well-written but see some minor wording suggestions below.
In general, the Methods section is not adequate. Many critical details are missing- see various comments below. At present the study could not be replicated by a reader of this paper.

Minor comments:
Line 43: “etc.” as a reference not appropriate- fill in missing citations.
45: Add citation for arthropod/unicellular size overlap. Many readers will be amazed by this.
49: See & cite recent papers by Eberhard & Wcislo on miniaturization in spiders.
51-52: Reword to “Some of these changes have evolved convergently in several taxa…”
54: Delete “and many more”
70-87: This section is overly detailed yet does not provide critical information for interpreting what is already known. Just listing species/tissues studied does not tell us what the gaps are in our knowledge. What do we not know yet that your data/study addresses?
It might be useful to turn this block of text into a table listing species by what tissues was studied. Then in the text of the paper, briefly describe any relevant patterns seen. What, if anything, do all these studies tell us about these very small-bodied animals? Even “big” collembola are still tiny!
89: Units needed! “only 0.4 long (Zimdars and Dunger, 1994).”
93: “analyze the effects of miniaturization.” Briefly state how you do this- by comparing with other collembola? Other arthropods?
105: More details on histology methods are needed: recipes/sources of reagents, or at least cite papers giving these methods in detail.
110: Again, more details needed in methods. Give exact names & versions of all software & hardware (cameras); what resolution were photos taken at? Illumination source? Etc. Make it possible for other researchers to replicate your methods exactly.
127: Critical- How were the measurements made? Need details in the Methods section!
Figures are almost all 3D reconstructions. It is important to describe carefully how they were made. How were different tissues identified?
137: “The cuticle thickness is 0.31–1.24 μm thick and longitudinal sections 0m (M = 0.57, n = 80).” How was this determined? In general- you must provide detailed explicit methods for how all data were collected throughout the Results.
202-349: I don’t see the value of naming every muscle in a primary literature paper about the evolution of miniaturization. This section can be reduced or deleted- or only mention if there are novel/surprising structures (or absences) that deviate from previously described arthropod or collembolan anatomy.

Experimental design

See above

Validity of the findings

See above

Additional comments

See above

·

Basic reporting

The text needs a thorough English language revision. There are several language ambiguities that should be corrected.

Throughout the Results: all measurements should be presented as actual measurements with variation, mean, and number of specimens measured, not just stating ‘about’ measurements. I would recommend adding a table with all these data and referring to it throughout the text.

Throughout the text: After referring to a species once with the full scientific name, refer to it with an abbreviated name, e.g. Anurida maritima -> A. maritima.

The authors should pay special attention to the readability of the text and how to streamline reading the text together with the figures. In a descriptive study which consists largely of lists of anatomical features the structure of the text and frequent references to the correct figure panels is essential, especially in places where several figures are needed to follow the text. Keep the reader in mind when writing!

In the chapter ‘Muscular system’ the different sections (Musculature of the head, the thorax, etc.) should be clearly separated and marked (with bold text or subheadings) to make it easier to read and follow.

Throughout the figures and figure legends: Make sure all abbreviations found in the figure are explained in the figure legend (except when it’s stated that they are in the main text), and that all of the terms explained in the legend are found in the figure.

Specific comments:

Lines 127-134 Description of general morphology: Most people associate collembolans with the furca, so you should describe the absence of it here. Also mention the absence of eyes here so it doesn’t come as a surprise in line 426.

Typing error in the Table S1 heading.

Experimental design

A minor weakness:
No mention of animal collection permits, or 'animals were handled according to all applicable laws' statement.

Validity of the findings

No comment.

Additional comments

This self-contained, descriptive study describes the anatomy of a minute collembolan species, constituting the first data to study miniaturizarion in non-insect arthropods. The research is done well and the results are a valuable addition to the existing body of knowledge. My main concerns are about the language and the presentation of the data, which, in my opinion, need major revision. Please see the annotated pdf of the manuscript for more details.

·

Basic reporting

This paper deals with the anatomical description and 3D reconstruction of the Collembola Mesaphorura sylvatica, a specially minute species, and comment on the effects of miniaturization on the anatomy.
The description is clear and comprehensive, and globally well-written. A small number of language corrections are need, that I annotated in the manuscript pdf.

The review of previous anatomical descriptions of Collembola seems comprehensive to me, and the works of major authors, old and recent, were rightfully cited. I however recommend to improve the introduction by providing more general background/context for readers:

1- being not familiar with the concept and background knowledge on miniaturization, I would have appreciated a more consistent introduction to it. The introduction never really give indications of size scales at which miniaturization is studied, and some biological facts should be provided to introduce the field of study. I made further comments in the text.

2- a three or four lines presentation of the studied species should be provided : taxonomy, ecology and morphology: aside of size, Collembola species differ tremendously in habitus, a part of this being reflected in the split into 4 orders. I made further comments in the text,

Article structure is good and professional. I think that the muscular system description could be formatted into a table rather than in a 3.5 pages of telegraphic style, as it seems to always repeat the same structure.

The authors did not mention whether the 3D models would be shared as supplementary material. I think it should.

Experimental design

Lack of previous complete anatomical reference for a Collembola species is a limitation to comparative anatomy. This work provides such a need reference, that will be extremely useful for further investigations. Methods of investigation comply with high standard in my opinion.

Validity of the findings

Factual description is well-done. I suggest to beef up a bit the interpretation part, and clearly state what can be said of size change impact on the organism functions (if possible).

Additional comments

This is a nice and needed work. A comprehensive framework to study miniaturization with Collembola as a model should include descriptions and comparisons of closely related small and large species throughout the main lineages. I hope this is within your prospects.

---

## Round 0.2 · Minor Revisions

Dear Dr. Panina and colleagues:

Thanks for revising your manuscript. The reviewers are very satisfied with your revision (as am I). Great! However, there are a few minor edits to make. Please address these ASAP so we may move towards acceptance of your work.

Best,

-joe

·

Basic reporting

The paper meets the standards established by the Journal.

Experimental design

The problem is clearly stated, original evidence has been obtained and clearly presented

Validity of the findings

Conclusion flows logically from data analysis and discussion.

Additional comments

In the revised version, the criticisms raised on the original submission have been satisfactorily addressed. Minor editorial suggestions are provided with the attached annotated files.

·

Basic reporting

Figures:

Generally all figure panels should be referred to in the main text. Here figures 1, 5 and 6 are mentioned as entities but not by panels. Fig. panels 2A, B, D and Fig. 3C are not mentioned in the text.

As I said in my previous review about the figures and legends: Make sure all abbreviations found in the figure are explained in the figure legend (except when it’s stated that they are in the main text), and that all of the terms explained in the legend are found in the figure.

Fig.1: explanation of “vl” is missing from the legend.

Fig. 3: explanation of “An” and “pst” is missing from the legend.

Fig. S1: The legend has the following items which are missing from the images: oes, ova, pst. The images have following items which are missing from the figure legend: oe, eg, sg, bp.

Table S2:

I found this table very useful! However, there are a couple of small things to correct:

skeleton volume is given in the table S2 as 0.048 but in the text (line 182) as 0.044 nl. Which one is correct?

To avoid confusion about the total volumes, please indicate clearly that the brain measurement is part of the CNS measurement, for example:
Central nervous system 0.051 6.3
of which the brain 0.016 2.2

Table S3:

typing error in the table legend.

Main text:

The rebuttal letter makes no mention of a language revision, but I find this version much easier to read and understand than the original one.

Line 189: “The slender pharynx is about 4.2 μm (M = 4.2, n = 8).” Is that the diameter?

L213: Typo “Itterminates”

L218: Typo “gangliashift”

L275: delete “with”

L308: Typo “beexplained”

L340: replace “what” with “which”


References:

L52: Reference to Eberhard & Wcislo, 2011 is missing from the reference list.

L570: Reference to Manton, 1964 is not mentioned in the text.

Experimental design

No comment

Validity of the findings

No comment

Additional comments

The manuscript has improved greatly from the first version! Especially its readability and the flow of the text is much better. I am also mostly happy with the way the authors have addressed my previous concerns. In my opinion, there are only a few minor fixes that need to be done before the manuscript is in a publishable shape.

---

## Round 0.3 · accepted · Accept

Dear Dr. Panina and colleagues:

Thanks for re-submitting your revised manuscript to PeerJ, and for addressing the concerns raised by the reviewers. I now believe that your manuscript is suitable for publication. Congratulations! I look forward to seeing this work in print, and I anticipate it being an important resource for research communities studying springtails.

Thanks again for choosing PeerJ to publish such important work.

-joe